# Method development and validation for rapid identification of epigallocatechin gallate using ultra-high performance liquid chromatography

Ramakrishna U. V.[1], Shyam Sunder R.[2], Rajesh Kumar K.[1], Sukesh Narayan Sinha[1]*

1 Food Safety Division, ICMR—National Institute of Nutrition, Tarnaka, Hyderabad, Telangana, India,
2 University College of Technology, Osmania University, Tarnaka, Hyderabad, Telangana, India

* sukeshnr_sinha@yahoo.com

**Data Availability Statement:** All relevant data are within the manuscript.

**Funding:** The authors received no specific funding for this work.

## Abstract

Although Epigallocatechin gallate (EGCG) is the most available and beneficial catechin found in tea, its auto-oxidation property may lead to toxicity when consumed in large quantities. Thus, there is a need to quantify the EGCG, which enables to study the pharmacological characteristics of the compound. The study aimed to develop and validate a rapid and accurate analytical method for quantitative determination of EGCG. Standard EGCG was used to conduct trials for the optimization of the analytical method using Ultra–High Performance Liquid Chromatography (UHPLC). Tests for validation (specificity, linearity, accuracy, system suitability, method precision, robustness, and ruggedness) were performed. The preliminary trials yielded an analytical method with good peak shape and acceptable system suitability which was further validated. The method was shown to be specific, with a linear correlation coefficient of > 0.9996 and accurate with acceptable recovery rate (99.1% to 100.4%). Acceptable system suitability and method precision were confirmed with a relative standard deviation (less than 2%). Further, robustness and ruggedness experiments also demonstrated the suitability of the present analytical method. The method developed for determination of EGCG was validated as per the International Council for Harmonisation of Technical Requirements for Pharmaceuticals for Human Use (ICH) guidelines and thus can be used in routine compliance tests in the laboratory for further studying/characterizing the properties of EGCG.

## Introduction

Epidemiological studies have reported the favourable benefits of tea consumption in varied population [1]. The benefits mainly can be attributed to the polyphenols, especially Epigallocatechin 3-gallate (EGCG), for its antioxidant properties [2]. EGCG is known to alleviate chronic diseases such as obesity, type-2 diabetes, lipid metabolism abnormity, atherosclerosis, cardiovascular diseases, and cancer when consumed in safe doses [3,4]. The pro-oxidation action of

**Competing interests:** The authors have declared that no competing interests exist.

EGCG is regarded as the crucial mechanism for its protective functions such as induction of adaptive responses and detoxifying capacities [5].

Intriguingly, studies have reported that EGCG in a higher dose can lead to health risks, majorly by inducing hepatotoxicity in both animals and human beings. The studies postulate that EGCG may also experience auto-oxidation, which leads to the generation of Reactive Oxygen Species (ROS) and thereby induce toxicity [6]. However, research is still being carried out to weigh the benefits and against toxicity to assess the safe physiological dose of EGCG. Therefore, this calls for assessing its pharmacological properties, which necessitates elucidating and comprehensively understanding the structure of EGCG. Few earlier studies across the literature, focused on ameliorating the potential side-effects of these compounds by various therapies and also how to alleviate these side effects by enhancing its bioavailability.

Previously, methods have been developed to estimate the EGCG using Spectrophotometry [7], High Performance Liquid Chromatography (HPLC) [8,9], Reverse Phase (RP) HPLC [10,11] and Ultra-Performance Liquid Chromatography (UPLC) [12,13], Ultra-Performance Liquid Chromatography-Tandem Mass Spectrometry (UPLC-MS/MS) [14], Ultra-Performance Liquid Chromatography/Electrospray Ionization–Mass Spectrometry (UPLC/ESI-MS), Ultra-Performance Liquid Chromatography-Diode Array Detector-Mass Spectroscopy (UPLC-DAD-MS) [15] and Ultra-Performance Liquid Chromatography—Time-of-Flight—Mass Spectrometry (UPLC-TOF-MS) [16]. Also keeping in view of the low bioavailability of the compound and the limited studies to identify EGCG alone from the samples, there is a strong need to develop and validate a simple and rapid method to identify EGCG.

Therefore, the study aimed to develop an analytical method to detect EGCG using a simple chromatography technique of UHPLC and also to validate the developed method by establishing the characteristic parameters and validation criteria were specificity, linearity, accuracy, precision (system suitability and method precision), robustness and ruggedness (intermediate precision). The acceptable criterion for each parameter was carried out under the guidelines given by the International Council for Harmonisation of Technical Requirements for Pharmaceuticals for Human Use (ICH). [17].

## Materials and methods

### 2.1 Instrumentation

Agilent 1290 Infinity II—UHPLC system (Agilent, California, US) was utilized for method development and validation. The instrument was provisioned with a High-Speed Pump, Multicolumn Thermostat (MCT) pump, an Ultra-Violet (UV) detector, an autosampler, and a control module. The chromatographic separation was executed on an AcquityC18 (50 mm x2.1 mm ID) 1.8μm of Waters (Milford, Massachusetts, USA). The software installed in the system was Openlab CDS Version 2.3.4 for data analysis and evaluation. An AB Sciex triple quadrupole mass spectrophotometer with analyst software was utilized for mass identification of the compound.

### 2.2 Chemicals and reagents

Pharmaceutical grade standards of EGCG (purity > 96%) were procured from M/s Cayman Chemicals Company (Ann Arbor, Michigan, US). Solvents such as methanol, acetonitrile, water of HPLC grade were purchased from M/s J.T.Baker Avanator (Radnor, Pennsylvania, USA), and potassium hydrogen phosphate ($K_2HPO_4$), potassium dihydrogen phosphate ($KH_2PO_4$) were procured from M/s Sigma Aldrich (St. Louis, Missouri, USA).

## 2.3 Chromatographic conditions

The mobile phase used for the study was phosphate buffer: methanol in the ratio of 70: 30 and was set in isocratic mode. The phosphate buffer solution was prepared by adding 3.394 gram of sodium phosphate monobasic and 20.209 gram of sodium phosphate dibasic to 800 mL of distilled water in a 1000 mL beaker. The pH was adjusted to 6.8 using HCl or NaOH and distilled water was used to make up the volume. Subsequently, the buffer solution was subjected to filtration using 0.45 μm membrane filter and then degassed by sonication for 15 minutes. The mobile phase was maintained at a flow rate of 0.5 mL/min. The chromatographic analysis was carried out on Agilent 1290 Infinity II—UPLC system, which was fitted with a UV detector set to 272 nm for obtaining chromatographic data. Waters Acquity C18 (50 mm x 2.1 mm ID) 1.8 μm Column was used for the chromatographic separation, which was maintained at 25°C. The injection volume was 20 μL. The total run time of the analytical method was set to 5 minutes in the UHPLC system.

## 2.4 Preparation of standard solution

The standard stock solution of EGCG (obtained from M/s Cayman chemicals) was prepared by dissolving 100 mg of EGCG in 100 mL of diluent (mobile phase). Subsequently, the filtration of the solution was done using a 0.45-micron syringe filter and sonication was done for 5 minutes before using for chromatographic analysis.

## 2.5 Extraction of EGCG from green tea

The procedure for extraction of EGCG from green tea is explained below in Fig 1. About 1 gram of green tea leaves (collected from the market) was dissolved in 10 mL methanol. Further, the solution was subjected to continuous vortex for 30 minutes and subsequently centrifuged at 2000 rpm (15 minutes), and the supernatant solution was collected for further processing. The supernatant was then extracted using QuEChERS (Quick, Easy, Cheap, Effective, Rugged and Safe) technique to remove the potential pigments from the green leaf solution. The obtained supernatant was taken into the Agilent Dispersive SPE 15mL [Fat + Pigments Association of Analytical Communities (AOAC)] tubes and was subjected to vortex for about 20 minutes. The tube was again centrifuged (2000 rpm, 10 minutes) to obtain the supernatant, which serves as the final solution. This solution was then filtered through a 0.2 micron syringe filter (Nylon) before analysis on the UHPLC instrument for the identification of EGCG.

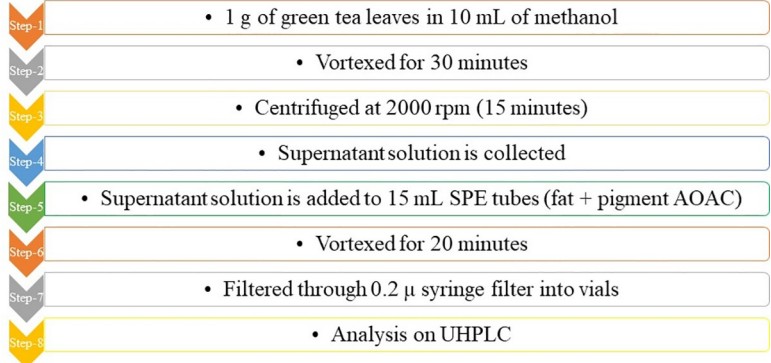

**Fig 1. Extraction Procedure of EGCG from green tea leaves.**

## 2.6 Method validation

**2.6.1 Specificity.** The specificity of the analytical method is its ability to differentiate between the analyte and the other substances in the sample matrix and thereby generate signals which are free from interference [18]. In a UHPLC method, it is confirmed by complete separation of the analyte peak from other peaks, which may originate from the sample matrix and also quantify the analyte in complex matrices. Evaluation of specificity was performed by injecting separately 10 μL of blank (mobile phase) and placebo (phosphate buffer solution) into the system.

**2.6.2 Mass Spectrometry (MS) analysis.** This analysis was performed on Mass Spectrometry (AB Sciex 4000 Q Trap) installed with Analyst software. The standard and sample extracted solutions (100μg/mL) were run to ensure that both the analytes were chemical identical with similar molecular structures. To obtain optimum sensitivity and selectivity, Electrospray Ionization (ESI) technique was operated in the positive ion mode which was used for MS/MS Multiple Reaction Monitoring (MRM) analysis. Further, the optimized compound parameters viz., Declustering Potential (DP), Collision Energy (CE), and Cell Exit Potential (CXP) were evaluated.

**2.6.3 Linearity.** Linearity was evaluated by carrying out the chromatographic analysis of several EGCG standard solutions with increasing concentrations and by establishing the calibration plot of the response vs. concentration, which visually approximates a straight line. Additionally, linear regression analysis was performed to assess the linearity of the calibration curve by employing the least square linear regression method to obtain the slope, intercept and correlation coefficient [18]. To determine the linearity of the evaluated method, standard solutions of EGCG stock solution was diluted in mobile phase, yielding standard solutions with concentrations of 50, 75, 100, 125 and 150 μg/mL. Subsequently, the solution was filtered using a 0.45micron syringe filter, which was later sonicated for 5 minutes before injecting into the UHPLC system to obtain the chromatograms. Further, linearity studies were performed for EGCG in green tea by spiking the matrix at different concentrations (20, 40, 60, 80, 100 μg/mL) and extracting the samples as mentioned above and subjected for UHPLC analysis.

**2.6.4 Accuracy.** The accuracy of the analytical method is determined as the nearness of the obtained value to the true value. In the present study, accuracy was determined by spiking the reference standards of the EGCG which were administered into the blank matrix, yielding solutions containing analyte with concentration levels of 50%, 100%, 150%. At every concentration level, about three samples were prepared and later analyzed. The recovery studies were performed in triplicates (for each sample), and the percentages of recovery and mean recovery were calculated for EGCG.

**2.6.5 Precision.** Precision is the degree of agreement/closeness among the results of each test when the method is subjected to multiple homogenous samples [18]. It can be explained as the reproducibility of measurement, and to determine the same, series of measurements (of similar quantities) were carried out. Thus, the system and method precision were evaluated using a standard solution of 100 μg/mL of EGCG, which was administered six times and the chromatograms were recorded for the same. Further, to determine the precision, statistical analysis such as Standard Deviation (SD) and the percent Relative Standard Deviation (% RSD) of variables such as Retention Time (RT), peak area, number of Theoretical Plates (TP) and Tailing Factors (TF) were measured for the test results of multiple aliquots.

**2.6.6 Limit of Detection (LOD).** The limit of detection (LOD) or detection limit (DL) can be defined as the lowest concentration of an analyte which can be reliably detected in the test sample. The LOD was calculated by taking 3.3 times of standard deviation of the response

(σ) and the slope of the calibration curve (S) as given in Eq 1.

$$LOD = \frac{3.3\,\sigma}{S}$$  Eq (1)

Where, σ = the standard deviation of the response
  S = the slope of the calibration curve

**2.6.7 Limit of Quantification (LOQ).**  The lowest concentration level at which a measurement is quantitatively meaningful is called the limit of quantitation (LOQ) or quantification limit (QL). This is most often defined as 10 times the signal-to-noise ratio. Thus, the LOQ is 10 times the standard deviation (σ) of the response and slope (S) of the calibration curve as given in Eq 2.

$$LOQ = \frac{10\,\sigma}{S}$$  Eq (2)

Where, σ = the standard deviation of the response
  S = the slope of the calibration curve

**2.6.8 Robustness.**  The robustness of the method determines the susceptibility of a method to small changes such as pH values, temperature, mobile phase composition, etc. which might occur during routine analysis at the laboratory [18]. In the present study, robustness was investigated by deliberately making the following alterations in the analytical method: (i) Flow rate: ± 0.1 mL/min, (ii) Column Temperature: ± 10˚C

At each condition, the standard solution of EGCG was administered into the chromatographic system in triplicates. The robustness of the method was assessed by calculating the % RSD of the peak area after three consecutive injections of the standard solution.

**2.6.9 Intermediate precision (ruggedness).**  Intermediate precision (ruggedness) is the measure of precision evaluated using pre-determined conditions: similar measurement procedure, similar measuring system, similar location, and replicate measurements on the same objects over a prolonged period of time. In the present study, the ruggedness was investigated using six individual sample preparations and % RSD was calculated.

## Results and discussion

### 3.1 Method development and optimization

Information on the physiochemical properties of EGCG was reviewed from the literature search. Further, the UHPLC method was developed to optimize the chromatographic conditions such as mobile phase, column, and wavelength for recording the chromatograms. Preliminary trials were carried out by altering the variables mentioned above and a set of chromatographic conditions were tested. Trail 1 yielded chromatogram with broad and asymmetric peaks; therefore, to minimize this, the acidity of the mobile phase was increased by adding glacial acetic acid as mentioned in trial 2. Additionally, in trial 2, only the column size was decreased as the retention time of the compound was around 2 minutes and all other specifications were maintained. However, this yielded chromatograms with split peaks, and hence in trial 3, Triethylamine was added. This also yielded unsatisfied results with multiple peaks. Further, to address the issue of multiple/splitting peaks, the buffer was introduced (trail 4) into the mobile phase, which usually helps to reduce the peak splitting and helps to maintain the pH of the solution. Since fronting of peaks was observed in this trial, the volume of buffer was increased in the mobile phase (trail 5). The results obtained from the preliminary optimization are given in Table 1 and it was observed that trial-5 was found to result in good peak shape, met the system suitability requirements, and thus was accepted and evaluated. The

**Table 1. Details of the preliminary trials and the results of the optimization.**

| | Mobile Phase composition | Column features | Flow Rate & Run Time | Observation | Result |
|---|---|---|---|---|---|
| **Trial 1** | Methanol: Water 70: 30 | Phenomenex C18 (150x2.1 mm ID) 1.8 μm | 0.5 mL/min 10 Min | • Broad peak and asymmetric factor do not meet the system suitability | Rejected |
| **Trial 2** | Water: Methanol: Glacial acetic acid 75: 25: 0.7 | Waters AcquityC18 (50 mm x2.1 mm ID) 1.8 μm | 0.5 mL/min 15 Min | • Peak splitting found and the resolution was too low • The asymmetry factor for EGCG does not meet the system suitability requirements | Rejected |
| **Trial 3** | Water: Methanol: Triethylamine 75: 25: 0.7 | Waters AcquityC18 (50 mm x2.1 mm ID) 1.8 μm | 0.5 mL/min 10 Min | • Multiple peaks were found, and the baseline was found to be unstable. • The efficiency of the method is very l ow | Rejected |
| **Trial 4** | Phosphate buffer: Methanol 30: 70 | Waters AcquityC18 (50 mm x2.1 mm ID) 1.8 μm | 0.5 mL/min 10 Min | • Fronting of peaks was observed and asymmetric factor does not meet the system suitability • The baseline was found to be unstable | Rejected |
| **Trial 5** | Phosphate buffer: Methanol 70: 30 | Waters AcquityC18 (50 mm x2.1 mm ID) 1.8 μm | 0.5 mL/min 8 Min | • All the system suitability requirements were met • The peak Asymmetry factor was less than 2 • The efficiency was more than 2000 | Accepted & Optimized |

chromatogram of EGCG obtained using the studied condition is given in Fig 2A. In the present study, the RT of EGCG peak was 2.1 minutes, which is found to be rapid as compared to previous studies in which RT was reported as 9 minutes and 5 minutes [10, 14].

The extracted solution from green tea was analysed on UHPLC under the selected conditions, and the identification of EGCG was obtained which can be seen in the chromatogram (Fig 2B).

## 3.2 Method validation

**3.2.1 Specificity.** The specificity of the method was determined by comparing the chromatograms attained from the blank (mobile phase) and placebo solution (phosphate buffer solution). For this purpose, analysis was conducted on placebo and the chromatogram is given in Fig 2C. The chromatograms of blank and placebo solutions were eluted before the peak of EGCG. It can also be noticed from the chromatograms that at the retention times of EGCG, no other peaks were found, confirming that the analyte was pure as there are no co-eluting peaks. Thus, it was observed that diluent or excipient peaks do not intervene with the EGCG peak, and this confirmed the specificity of the method developed in the present study.

**3.2.2 Mass Spectrometry (MS) analysis.** The mass of the standard EGCG was found to be 459.0. Using similar conditions, the sample extracted from green tea was also analyzed and the chromatograph is given in Fig 3, the mass was also found to be 459.0 which is similar to that of EGCG. Further, the optimized compound parameters viz., DP, CE and CXP were found to be 66 volts, 17 volts and 8 volts respectively.

**3.2.3 Linearity.** The standard stock solutions of EGCG were prepared and diluted to obtain five concentrations of 50, 75, 100, 125 and 150 μg/mL. The prepared solutions of EGCG were injected into the system, and the chromatograms were recorded. Peak areas of the analyte were obtained for each corresponding concentration and noted. A graph was plotted for EGCG against the concentrations of the solutions and the peak areas (Fig 4A). From the regression analysis, the linear equation obtained was y = 7.0638x - 169.54 and the correlation coefficient ($R^2$) was $\geq$ 0.9991 for EGCG, indicating a linear relationship between concentration of the analyte and the area under the peak. Thus, the present analytical method was found to be linear in the specified range.

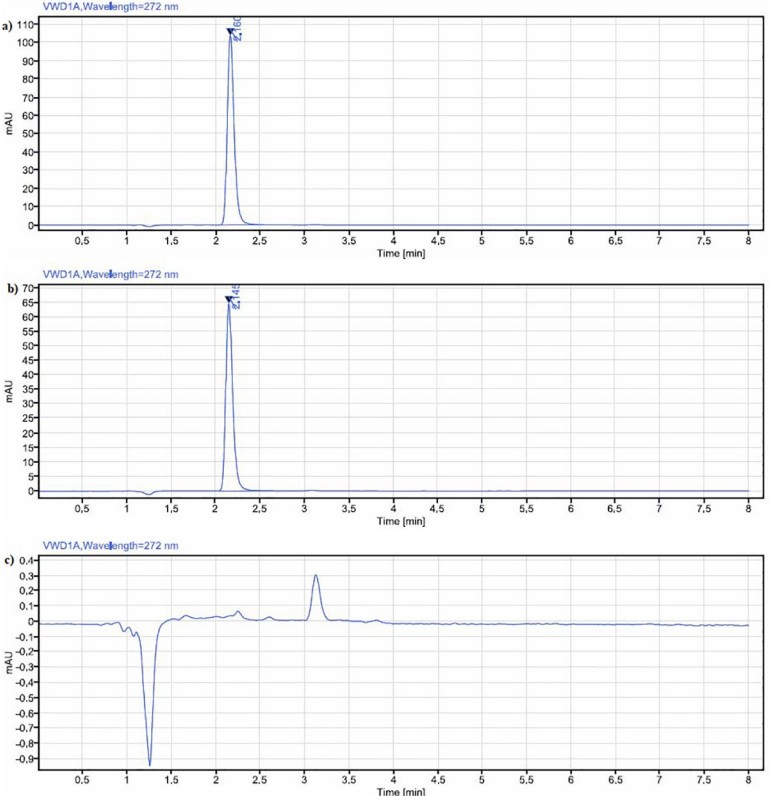

**Fig 2.** A. Chromatogram of EGCG by the developed method. B. Chromatogram of EGCG identification in green tea leaves matrices. C. Chromatogram of Placebo.

Similarly, different concentrations of EGCG (20, 40, 60, 80, 100 μg/mL) were spiked into various aliquots of green tea, and the matrix was extracted to study the linearity. Chromatograms were obtained using UHPLC, and the graph was plotted between the concentrations and the peak areas (Fig 4B). The satisfactory correlation coefficient ($R^2$) of ≥ 0.9988 was obtained using the linear equation y = 20.797x + 47.83, which indicates the linear relationship between the concentrations obtained from the green tea matrix and the areas.

**3.2.4 Accuracy.** In the present study, recovery studies are performed to check the accuracy of the method. The reference standards of the EGCG were made at the concentration levels of 50%, 100%, 150%. The studies for recovery were performed; percentage recovery and percentage mean recovery were calculated for EGCG (Table 2). The results of the recovery studies observed the recovery rate from 99.1% to 100.4% at all three levels. The results were within the acceptable criteria for recovery studies for an analytical method.

**3.2.5 System suitability and method precision.** To certify that the analytical system is running correctly and can report accurate and precise, results were assessed by injecting samples of EGCG (100 μg/mL) to record the chromatograms. The % RSD of retention time (RT) was found to be 0.1% and the % RSD for peak area was 0.1%. While the number of theoretical plates (TP) was higher than 30000 (with a range of 30521 to 30581) for all analyte peaks, the tailing factors (TF) were all less than 2.0% (with a range of 1.22 to 1.27).

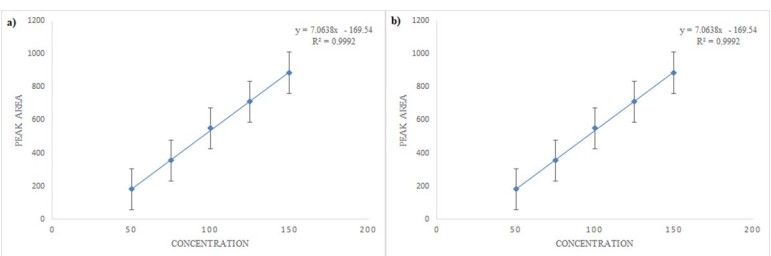

**Fig 3. Mass Spectra of EGCG in green tea.**

Further, the method precision was obtained by injecting sample solutions of EGCG six times at concentration (100 μg/mL), prepared separately. The chromatograms were reported and the results are as follows. The % RSD of assay for 6 samples determinations of EGCG was

**Fig 4.** A. Calibration curve obtained between peak area and concentration obtained for standard EGCG solutions upon linearity determination. B. Calibration curve obtained between peak area and concentration obtained for EGCG solutions from green tea upon linearity determination.

**Table 2. Results for determination of recovery of the analytical method.**

| Spiked weight of Standard (mg) | Replicate number | Concentration of the solution (µg/mL) | Area | Concentration recovered (µg/mL) | Recovery (%) |
|---|---|---|---|---|---|
| 50 | 1 | 50.0 | 281.9 | 50.1 | 100.1 |
| | 2 | 50.0 | 281.6 | 50.0 | 100.0 |
| | 3 | 50.0 | 281.8 | 50.0 | 100.1 |
| 100 | 1 | 100.0 | 564.4 | 100.2 | 100.2 |
| | 2 | 100.0 | 564.5 | 100.3 | 100.3 |
| | 3 | 100.0 | 565.6 | 100.4 | 100.4 |
| 150 | 1 | 150.0 | 840.2 | 149.2 | 99.5 |
| | 2 | 150.0 | 837.2 | 148.7 | 99.1 |
| | 3 | 150.0 | 837.3 | 148.7 | 99.1 |

found to be 0.1% and the % assay was also within the limits (95 to 105). Since the results obtained of both system and method precision showed that the variables obtained were within the acceptable limits, it can be considered that the system and method is precise.

**3.2.6 LOD and LOQ.** The LOD and LOQ were determined based on the signal to noise ratio of 3.3 and 10, respectively. The standard deviation (σ) was found to be 0.364, and the slope (S) of the calibration curve (obtained from the linearity) was observed to be 7.06. Therefore, the LOD was calculated as 0.170 µg/mL, and the LOQ was 0.515 µg/mL. The limits in the present study were found to be lower than the previous study, which reports LOD as 0.528 µg/mL and LOQ as 1.600 µg/mL, thus suggesting the present method was more sensitive [13]. Similarly, the LOD and LOQ with regard to the green tea leaf extract were calculated. The standard deviation (σ) was obtained as 0.364 and slope (S) as 20.797 (from the calibration curve of the green leaf extract samples). The LOD and LOQ were found to be 0.05 µg/mL and 0.17 µg/mL respectively.

**3.2.7 Robustness.** The Robustness of the investigated analytical method was tested to evaluate the impact of minor changes in the U-HPLC system conditions on parameters like system suitability of the new method. The results due to deliberate variation in the method conditions are summarized below in Table 3, which includes minor change of flow rate and column temperature. For all the modifications and the three consecutive injections, the % RSD values for RT, Tailing Factor, Theoretical Plates, and peak areas were determined. The results obtained showed that the minor modifications applied in the robustness test had no significant change, and the variables determined in the test were found to be within the acceptable limits.

**3.2.8 Intermediate precision (ruggedness).** The measure of intermediate precision/ruggedness was performed using six preparations of the EGCG individually into the chromatography system to determine the % assay of individual samples. The average % assay was found to be 99.0% with a range of 97.7% to 99.4 and the RSD (%) for six preparations assay values was 0.7. From the above results, it can be observed that the SD of % assay was 0.67 and RSD (%) was 0.7, which were within the acceptance criteria, indicating that the present method is rugged.

## Conclusion

This study was carried out to develop and evaluate a method for the estimation of EGCG using UHPLC. The results of the validation study deduced that this method was simple, rapid, accurate, precise, robust and rugged. The method was validated in accordance with the guidelines

**Table 3. Results for determination of Robustness of the analytical method.**

| Chromatographic changes | | Retention Time (min) | Peak Area | Tailing Factor | Theoretical Plates |
|---|---|---|---|---|---|
| Flow rate (mL/min) | 0.4 | 2.820 | 746.77 | 1.32 | 39481.87 |
| | | 2.826 | 746.00 | 1.30 | 39439.75 |
| | | 2.827 | 746.27 | 1.33 | 39190.81 |
| | Mean | 2.824 | 746.35 | 1.31 | 39370.81 |
| | SD | 0.003 | 0.3906 | 0.015 | 157.3 |
| | RSD (%) (n = 3) | 0.1 | 0.05 | 1.1 | 0.3 |
| | 0.6 | 1.705 | 449.05 | 1.22 | 27869.63 |
| | | 1.702 | 449.43 | 1.20 | 27829.76 |
| | | 1.7 | 449.31 | 1.24 | 27700.70 |
| | Mean | 1.702 | 449.26 | 1.22 | 27800.03 |
| | SD | 0.002 | 0.194 | 0.02 | 88.301 |
| | RSD (%) (n = 3) | 0.1 | 0.04 | 1.6 | 0.3 |
| Column Temperature (°C) | 25°C | 2.400 | 562.20 | 1.24 | 30348.93 |
| | | 2.404 | 562.16 | 1.27 | 3015.96 |
| | | 2.405 | 562.09 | 1.24 | 30255.85 |
| | Mean | 2.403 | 562.15 | 1.25 | 30372.58 |
| | SD | 0.002 | 0.055 | 0.017 | 130.17 |
| | RSD (%) (n = 3) | 0.1 | 0.009 | 1.3 | 0.4 |
| | 35°C | 1.937 | 615.62 | 1.28 | 32980.61 |
| | | 1.913 | 618.37 | 1.26 | 33285.38 |
| | | 1.899 | 616.53 | 1.28 | 33197.77 |
| | Mean | 1.9163 | 616.84 | 1.273 | 33154.59 |
| | SD | 0.019 | 1.400 | 0.011 | 156.90 |
| | RSD (%) (n = 3) | 1.002 | 0.2 | 0.9 | 0.4 |

laid down by ICH and found to be acceptable. The method could provide accurate and precise quantitative results under minor changes of chromatographic conditions.

## Acknowledgments

The authors are grateful to the Indian Council of Medical Research (ICMR) for granting the fellowship, ICMR-National Institute of Nutrition and Osmania University for the facilities, encouragement and support. The author would also like to specially thanks to Dr. Subba Rao G, Dr. M. Srujana and Mr. Sai Prasad, Chandra labs for their constant support.

## Author Contributions

**Conceptualization:** Ramakrishna U. V.

**Data curation:** Ramakrishna U. V.

**Formal analysis:** Ramakrishna U. V., Shyam Sunder R., Rajesh Kumar K.

**Investigation:** Ramakrishna U. V., Shyam Sunder R., Sukesh Narayan Sinha.

**Methodology:** Ramakrishna U. V., Rajesh Kumar K.

**Project administration:** Ramakrishna U. V.

**Resources:** Sukesh Narayan Sinha.

**Supervision:** Shyam Sunder R., Sukesh Narayan Sinha.

**Validation:** Ramakrishna U. V.

**Writing – original draft:** Ramakrishna U. V.

**Writing – review & editing:** Ramakrishna U. V., Shyam Sunder R., Sukesh Narayan Sinha.

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
