## [Decision Letter · Decision Letter 0]

29 Oct 2019

PONE-D-19-27318

Method Development and Validation for rapid identification of Epigallocatechin Gallate using Ultra-Performance Liquid Chromatography

PLOS ONE

Dear Dr. Sinha,

Thank you for submitting your manuscript to PLOS ONE. After careful consideration, we feel that it has merit but does not fully meet PLOS ONE’s publication criteria as it currently stands. Therefore, we invite you to submit a revised version of the manuscript that addresses the points raised during the review process.

We would appreciate receiving your revised manuscript by Dec 13 2019 11:59PM. To enhance the reproducibility of your results, we recommend that if applicable you deposit your laboratory protocols in protocols.io, where a protocol can be assigned its own identifier (DOI) such that it can be cited independently in the future. For instructions see: http://journals.plos.org/plosone/s/submission-guidelines#loc-laboratory-protocols

We look forward to receiving your revised manuscript.

Kind regards,

Marina Pinheiro

Academic Editor

PLOS ONE

**Journal Requirements:**

2. Thank you for stating the following financial disclosure: “No”   a) Please provide an amended Funding Statement that declares *all* the funding or sources of support received during this specific study (whether external or internal to your organization) as detailed online in our guide for authors at http://journals.plos.org/plosone/s/submit-now.     b)  Please state what role the funders took in the study.  If any authors received a salary from any of your funders, please state which authors and which funder. If the funders had no role, please state: "The funders had no role in study design, data collection and analysis, decision to publish, or preparation of the manuscript."

c) If the study was unfunded, please state "The author(s) received no specific funding for this work."

**Comments to the Author**

1. Is the manuscript technically sound, and do the data support the conclusions?

Reviewer #1: No

Reviewer #2: Partly

2. Has the statistical analysis been performed appropriately and rigorously? 

Reviewer #1: Yes

Reviewer #2: I Don't Know

3. Have the authors made all data underlying the findings in their manuscript fully available?

Reviewer #1: Yes

Reviewer #2: Yes

4. Is the manuscript presented in an intelligible fashion and written in standard English?

Reviewer #1: No

Reviewer #2: No

5. Review Comments to the Author

Reviewer #1: This manuscript describes the method development and validation for EGCG using UHPLC. The goals of the study are straightforward. However, there are several issues with the manuscript that need to be addressed. These points are listed below in order of appearance in the manuscript.

Page 2, abstract: The abstract needs to be rewritten and proofread for grammar. The are issues with the first two sentences. "Specificity" is list twice in line 7.

Page 2, abstract and throughout the paper. UPLC should be replaced by UHPLC as it is the proper term. UPLC is a trademark of Waters Corp.

Page 2, line 3 of introduction: EGCG is know to help alleviate some symptoms, but does not prevent all the diseases mentioned here.

Page 6, end of specificity section: The contents of the blank and placebo solutions must be described.

Page 7 and 8: Equations should be numbered.

Page 8, Robustness section: The authors state the temperature is varied +/- 10 degrees. What temperature is varied? Is it the cinematographic column temperature? the solution temperature? It needs to be more specific.

Page 9, Table 1: "The efficiency was very less." does not make sense. What efficiency is being described? Efficiency is not described as a criterion in the Methods section.

Page 10, Figure 2: All the chromatograms should have the same time axis some that the comparison is easier to see by the reader.

Page 11, top of the page: The difference observed by mass spectrometry must be explained. A difference of 0.8 Da is significant. What is the cause of the difference?

Page 13, LOD and LOQ section: The slope used in the calculation is form the standard curve. The slope from the tea leaves is different and should be used here if the authors are showing how their method works in leaves. Or both analyses should be presented. The last sentence on this page does not agree with the authors calculations presented and are the same as reference 13.

Reviewer #2: An Ultra-Performance Liquid Chromatography method for single-analyte quantification of Epigallocatechin gallate (ECGC) in green tea was developed. The idea of the work is clear and the method was properly validated, as it seems. However, there are several language inconsistencies and terminology mistakes in some points that must be clarified and corrected for a better understanding and evaluation of the scientific content. Hence, I would consider the paper for publication in PLOSONE provided that the authors conduct major revisions on the manuscript, as detailed in the pdf file attached. Needless to say, that a major revision of the English and the analytical language is highly advised for acceptance of the final manuscript.

6. PLOS authors have the option to publish the peer review history of their article (what does this mean?). If published, this will include your full peer review and any attached files.

Reviewer #1: No

Reviewer #2: No

---

## [Author Response · Author response to Decision Letter 0]

21 Nov 2019

Review Comments to the Author

Reviewer #1: 

Comment 1: Page 2, abstract: The abstract needs to be rewritten and proofread for grammar. The are issues with the first two sentences. "Specificity" is list twice in line 7.

Response: As per the reviewer’s suggestions, necessary corrections were made in the abstract (Page number 2). 

Comment 2: Page 2, abstract and throughout the paper. UPLC should be replaced by UHPLC as it is the proper term. UPLC is a trademark of Waters Corp.

Response: The term UPLC has been replaced by UHPLC throughout the revised manuscript. 

Comment 3: Page 2, line 3 of introduction: EGCG is know to help alleviate some symptoms, but does not prevent all the diseases mentioned here.

Response: As suggested, appropriate word has been replaced in the revised manuscript. 

Comment4: Page 6, end of specificity section: The contents of the blank and placebo solutions must be described.

Response: As desired, the contents of both blank and placebo solutions are described in the revised manuscript (Page numbers 6 and 11).

Comment 5: Page 7 and 8: Equations should be numbered.

Response: The equations given in page 8 are duly numbered in the revised manuscript. 

Comment 6: Page 8, Robustness section: The authors state the temperature is varied +/- 10 degrees. What temperature is varied? Is it the cinematographic column temperature? the solution temperature? It needs to be more specific.

Response: The temperature of the column was varied to evaluate the robustness of the developed method. The same is now incorporated in the revised manuscript (Page numbers 9 and 15).

Comment 7: Page 9, Table 1: "The efficiency was very less." does not make sense. What efficiency is being described? Efficiency is not described as a criterion in the Methods section.

Response: The efficiency of a chromatographic peak is a measure of the dispersion of the analyte band as it travels through the UHPLC system and column. Ideally, the chromatographic peaks would be pencil thin lines; however, due to dispersion effects the peaks take on their familiar ‘Guassian’ shape. Therefore, in this context, efficiency of chromatograms means asymmetric and unacceptable shape along with split in peaks and disturbance in the baseline which was unstable.

Comment 8: Page 10, Figure 2: All the chromatograms should have the same time axis some that the comparison is easier to see by the reader.

Response: In Figure 2, since the time of identification was found to be around 2 minutes, run time was decreased to 5 minutes instead of 8 minutes and hence the difference in the time scale was observed. 

Comment 9: Page 11, top of the page: The difference observed by mass spectrometry must be explained. A difference of 0.8 Da is significant. What is the cause of the difference?

Response: Generally, in mass spectra molecular ion is obtained as protonated and deprotonated ions. And deprotonation may have been observed in the green tea. We have used 4000 Qtrap which is not high-resolution MS.

Comment 10: Page 13, LOD and LOQ section: The slope used in the calculation is form the standard curve. The slope from the tea leaves is different and should be used here if the authors are showing how their method works in leaves. Or both analyses should be presented. The last sentence on this page does not agree with the authors calculations presented and are the same as reference 13.

Response: Since the aim of the study was to develop a method and validate a method for identification of EGCG, considering the slope from the standard curve obtained between peak area and concentration of standard EGCG solutions to calculate LOD and LOQ is sufficient and serves the purpose. Similarly, the LOD and LOQ with regard to the green tea leaf extract were calculated using the σ as 0.364 and S as 20.797 (obtained from the calibration curve of green tea leaf extract) and was found to be 0.05 and 0.17 respectively and is now incorporated in the revised manuscript (Page number 14).

Reviewer #2:

An Ultra-Performance Liquid Chromatography method for single-analyte quantification of Epigallocatechin gallate (ECGC) in green tea was developed. The idea of the work is clear and the method was properly validated, as it seems. However, there are several language inconsistencies and terminology mistakes in some points that must be clarified and corrected for a better understanding and evaluation of the scientific content. Hence, I would consider the paper for publication in PLOSONE provided that the authors conduct major revisions on the manuscript, as detailed in the pdf file attached. Needless to say, that a major revision of the English and the analytical language is highly advised for acceptance of the final manuscript.

Response: The revised manuscript was proof-read by a native English speaker and then crosschecked using Grammarly software for any grammatical and technical errors in the language.

Major Revisions

Question 1: Please revise the short title for “UPLC determination of EGCG” as quantitative analysis is presented in the manuscript.

Response: As per the suggestion of the reviewer, the short title is edited. 

Question 2: Please revise all units according to the International System and make them consistent across the entire manuscript text and figures (especially figure 1). All units must be separated from values by a space, except for percentage in which the number must be close to the value. Beware that “1 gr” and “micron” are not the correct way of expressing units (Figure 1 and respective manuscript text). In this sense, make the formatting consistent and correct throughout the whole manuscript and the figures. Please include values of n in brackets, as the number of determinations, whenever a relative standard value is given. 

Response: In the revised manuscript, all the units (both in figure and manuscript) are revised. Further, the number of determinations (n) is also included at the appropriate places. 

Question 3: Authors seemed to have been rather careless about significant figures. All inaccuracies must be corrected. For instance, LOD and LOQ should bare a single or two significant figures at the most.

Response: The method employed to calculate the LOD and LOQ values was by using the standard deviation and slope of the calibration curve. Since, these values were already given while representing the linearity in Figure 4a, additional figures were avoided. 

Question 4: Please revise the term “therapeutic doses of EGCG” (first paragraph of the introduction section) and clarify the sentence that follows “EGCG’s pro-oxidation action is regarded as a mechanism for its protective functions including its antioxidant properties”. This presents to me as an inconsistency to me as dose distinguished pro and antioxidant actions.

Response: As per the suggestion, the term and the sentence mentioned by the reviewer were edited (Page numbers 2 and 3). 

Question 4: A more solid explanation must be given concerning the purpose of developing of a single-analyte method for EGCG detection by chromatography. Accordingly, authors must revise the third paragraph of the introduction section and link it with the first sentence of the following, on page 3: “Despite previous studies, there is a need to optimize a simple and rapid method to identify the EGCG compound using a simple chromatography technique using UPLC system”. Also, remove “compound” and “using UPLC system” from the sentence. 

Response: The purpose of the present study was edited and the appropriate changes suggested by the reviewers were made (Page number 3). Also, the terms “compound” and “using UPLC system” were removed in the revised manuscript. 

Question 5: Still on page 3 (last paragraph), please write “to validate the method by establishing the characteristic analytical parameters” and “validation criteria were”. Also, the ICH guideline that has been applied must be referenced here. 

Response: The appropriate changes in terms of re-phrasing were made (Page number 3). The ICH guidelines which was considered in the present study was also referred (Reference number 17). 

Question 6: In section 2.2, clarify what do authors mean by “pharmaceutical grade samples of EGCG”. Do they refer to calibration standards? In section 2.4, please clarify what the “diluent” was. 

Response: Pharmaceutical grade samples of EGCG was corrected to Pharmaceutical Grade Standards (Page number 4). Mobile phase was used as the “diluent” and is mentioned in the revised manuscript (Page number 5).

Question 7: Before chromatographic analysis, a three-step sample preparation was employed. Please explain the need for this cumbersome procedure to extract EGCG. Also, I am afraid that some of the analyte may be lost in the filter immediately before analysis. Recovery percentages are quite high for such a multi-step procedure. Could the authors address this issue and clarify? Also, please add information on the syringe filter material.

Response: Most polyphenols including EGCG are soluble in solvents like methanol, water etc. Hence, green tea was dissolved in methanol to extract EGCG. Further, since the green tea is rich in pigments and may interfere/block the column, therefore to isolate these pigments SPE tubes were used. Additionally, to separate any undissolved large particles, filtration was done using 0.2 µ syringe filters (Nylon). 

Question 8: In section 2.6.2, please re-phrase”: Linearity was evaluated by carrying out the chromatographic analysis of several EGCG standard solutions with increasing concentrations and by establishing the calibration plot of the response vs. concentration, which visually approximates a straight line.” Also, “To determine linearity (…), EGCG stock solution was diluted in mobile phase, yielding standard solutions with concentrations of (…).”. 

Response: The suggested re-phrasing was done in the revised manuscript (Page numbers 6 and 7). 

Question 9: In section 2.6.3, please restrict the description to the strategy that has been used in this work for accuracy assessment. It is rather confusing how this determination was performed. Please re-phrase starting by “In the present study, accuracy was determined by spiking (…)”. 

Response: As per the reviewer’s suggestion, the phrase was revised and incorporated (Page number 7). 

Question 10: In section 2.6.5, please re-phrase as follows “the LOD is defined as the lowest concentration of analyte which can be reliably detected in the test sample.”

Response: The sentence re-phrasing was done as per the reviewer’s suggestion (Page number 8). 

Question 11: Please replace the word “optimise” by “studied”, “evaluated” or “investigated” whenever appropriate as no design of experiments was undertaken.

Response: As suggested by the reviewer, the term “optimise” was revised to appropriate word and included in the revised manuscript. 

Question 12: In section 3.1, replace “five such trials were conducted” by “set of chromatographic conditions were tested”. The authors must elaborate further on the options taken to establish the five trials as a poor discussion is provided. For instance, many variables have been changed from trial 1 to trial 2, preventing the isolation of the motive for rejection. Was it the column length of the mobile phase composition? Also, in table 1, please write “Mobile phase composition” and “Column features” in the appropriate column. What do authors mean by “Baseline is not proper”? Was it unstable, too much noise?

Response: As suggested by the reviewers, the selected were replaced in the section 3.1 (Page number 9). Further, elaborate discussion is now provided on the five trials taken up during the method development (Page numbers 9 and 10). The appropriate words were revised in table 1 (Page numbers 10 and 11). Further, as suggested by the reviewers, the terms were revised in the appropriate column of the table 1 as ‘baseline was found to be unstable’.

Question 13: Please eliminate section 3.2 and include comments in section 3.1. Renumber the rest of the section, accordingly.

Response: As per the suggestion of the reviewer, section 3.2 was eliminated and the rest of the sections were renumbered accordingly. 

Question 14: Concerning section 3.3.1, define “blank” and “placebo” appropriately. Please re-phrase the second sentence, as the idea is simply unclear.

Response: In the present, mobile phase was used “blank” and the phosphate buffer solution was used as the placebo. These are defined in the manuscript (Page numbers 6 and 11). Further, the second sentence has now been re-phrased in the revised manuscript. 

Question 15: Please explain the purpose of performing a mass spectrometry analysis (page 11). Also, I do not recall reading the specifics of the determination in the methods section, particularly the m/z values of the main transitions used for EGCG identification. Please add this information wherever appropriate.

Response: The primary function of mass spectrometry is to detect a chemical compound based on its mass-to-charge (m/z) ratio. In the present study, standard EGCG was initially run on mass spectrometry to identify its mass. Later, sample extracted from the green tea leaves was also run on mass spectrometry to identify its mass and to correlate the same with the standard. This ensures that both the analytes have chemical identity and structure of molecules. The purpose of performing this analysis was to confirm that both of them are similar chemical compounds. Further, the details of the mass spectrometry analysis are now added in the methods section (Page number 6). 

Question 16: The first three sentence of section 3.3.2 is just a repetition of the information of the respective experimental section. Please re-phrase them appropriately as matter of discussion. Please write R2 ≥ 0.9998. Also, in Figure 4a caption, re-phrase as “Calibration curve obtained between peak area and concentration obtained for standard EGCG solutions upon linearity determination.” The same phrasing should be applied for figure 4b. Please add error bars to the figures and the uncertainty obtained the slope and the intercept values. Finally, authors should explain the criteria to select the test concentration in light of the biological contents of EGCG. 

Response: The sentences mentioned by the reviewer were re-phrased. Further, “≥” was added at the appropriate places. The captions for Figures 4a and 4b were changed according to the suggestions and error bars were added to the Figures 4a and 4b. With regard to the criteria for selecting the test concentrations, since the working linearity range is predefined by the purpose of the method and it is known that concentration of EGCG is around 30-40% in green tea, we have considered the linearity range from the lowest concentration of 20 µg/mL. 

Question 17: Regarding Robustness, I do not see how temperature effect was evaluated. Please clarify. 

Response: For the purpose of the determination of robustness, the column temperature was changed and evaluated (mentioned in page numbers 9 and 15 of the revised manuscript). 

Question 18: Re-write the first sentence 3.3.4. it is just cumbersome.

Response: The sentence mentioned by the reviewer was revised and incorporated (Page number 14). 

Minor revisions:

Question 1: Please revise the whole text for typos and make sure the references are in the correct format and numbered throughout the text.

Response: The whole manuscript was thoroughly checked for typos and the references were edited and numbered through the text. 

Question 2: Please replace “same developed conditions” by “selected conditions” on page 10 (section 3.2) and whenever appropriate.

Response: The necessary corrections were made and incorporated (Page number 11).

Question 3: Abstract: Please correct “its auto-oxidation property may lead to toxicity”, write “linear correlation coefficient > 0.9996”, and remove average recovery. 

Response: The appropriate changes mentioned by the reviewer with regard to the abstract were made (Page number 2). 

Question 4: Remove capital letters from buffer salts, reagents, laboratory material (except for brands or equipment models), and analytical parameters within sentences. 

Response: The suggested changes were made and incorporated (Page numbers 4 and 5).

Question 5: Figures and Tables: - In figure 1, remove all capital letters within the sentences and the respective caption. Add the meaning of AOAC to the respective caption and to the manuscript text. Correct “supernate” to “supernatant”, also in the main text.

- Erase the column for average recovery on Table 2. Please write “Recovery (%), recovery values, and RSD (%)” in all tables instead of % Recovery, percentage recovery, and % RSD respectively.

- Please revise significant figures in Table 3 as those are not consistent and in agreement with method precision. Also, express RSD values with a single decimal figure in all cases. 

Response: The corrections mentioned with respect to the Figure 1 were duly made and incorporated in the revised manuscript. Table 2 and 3 were revised as suggested by the reviewer.

---

## [Decision Letter · Decision Letter 1]

2 Dec 2019

PONE-D-19-27318R1

Method Development and Validation for rapid identification of Epigallocatechin Gallate using Ultra-High Performance Liquid Chromatography

PLOS ONE

Dear Dr. Sinha,

Thank you for submitting your manuscript to PLOS ONE. After careful consideration, we feel that it has merit but does not fully meet PLOS ONE’s publication criteria as it currently stands. Therefore, we invite you to submit a revised version of the manuscript that addresses the points raised during the review process.

We would appreciate receiving your revised manuscript by Jan 16 2020 11:59PM. To enhance the reproducibility of your results, we recommend that if applicable you deposit your laboratory protocols in protocols.io, where a protocol can be assigned its own identifier (DOI) such that it can be cited independently in the future. For instructions see: http://journals.plos.org/plosone/s/submission-guidelines#loc-laboratory-protocols

We look forward to receiving your revised manuscript.

Kind regards,

Marina Pinheiro

Academic Editor

PLOS ONE

Reviewers' comments:

Reviewer's Responses to Questions

**Comments to the Author**

1. If the authors have adequately addressed your comments raised in a previous round of review and you feel that this manuscript is now acceptable for publication, you may indicate that here to bypass the “Comments to the Author” section, enter your conflict of interest statement in the “Confidential to Editor” section, and submit your "Accept" recommendation.

Reviewer #1: (No Response)

2. Is the manuscript technically sound, and do the data support the conclusions?

Reviewer #1: Partly

3. Has the statistical analysis been performed appropriately and rigorously? 

Reviewer #1: Yes

4. Have the authors made all data underlying the findings in their manuscript fully available?

Reviewer #1: Yes

5. Is the manuscript presented in an intelligible fashion and written in standard English?

Reviewer #1: No

6. Review Comments to the Author

Reviewer #1: This manuscript is improved over the previous version. However, the authors still have several issues that they need to address. The concerns are listed below.

1. The third line of the abstract is worded poorly and needs to be rewritten. Many of the previous wording concerns have been addressed, but a thorough proofreading is needed in any revision.

2. From the previous review "Comment 9: Page 11, top of the page: The difference observed by mass spectrometry must be explained. A difference of 0.8 Da is significant. What is the cause of the

difference?"

Response: Generally, in mass spectra molecular ion is obtained as protonated and

deprotonated ions. And deprotonation may have been observed in the green tea. We

have used 4000 Qtrap which is not high-resolution MS.

This response is not adequate. A difference of 0.8 Da is easily observed on a QTrap 4000. (I agree a QTrap 4000 is not a high resolution instrument.) If the authors think there is a protanation/deprotanation reaction, it needs to be demonstrated and explained. Why is there a Dalton shift between standard solutions and real samples?

3. Figure 2: All the chromatograms should have the same time axis so that the comparison is easier to see by the reader. I understand the change in run times, but the data should all have the same time scale.

4. Page 14, Section 3.2.6: The last part of this section is not clearly written. Further the calculated LOQ and LOD need to have units with the values.

5. There are still issues with significant figures in Tables 2 and 3. For example in Table 2, the concentration of a solution is listed as 50 ug/mL (1 significant figure), yet the recovered amount is listed as 50.06 ug/mL (4 significant figures). The recovered amount can not have more significant figures than the amount you started with, 1 significant figure. In Table 3, the number of theoretical plates is given to 0.01, this is beyond the precision of the measurement. Both tables need to be revised further.

7. PLOS authors have the option to publish the peer review history of their article (what does this mean?). If published, this will include your full peer review and any attached files.

Reviewer #1: No

---

## [Author Response · Author response to Decision Letter 1]

20 Dec 2019

Review Comments to the Author

Reviewer #1:

Response 1: As has been suggested by the reviewer, the third line of the abstract has been rewritten (Page number 2) and in-depth proofreading of the whole manuscript was done in the revised manuscript. 

Response 2: The authors are thankful to the reviewer for the comment. We have repeated the MS analysis for green tea extraction sample for the identification of EGCG several times and the mass was found to be 459.0 which is similar to mass of the standard EGCG which was also found at 459.0. Hence we could confirm that the developed method is suitable for identification of EGCG (Page number 11). 

Response 3: As suggested by the reviewer, all chromatograms were given with the same time axis (Figure 2a, 2b and 2c). 

Response 4: As suggested by the reviewer, the last part of the section 3.2.6 has been rewritten and also units were given along with the LOD and LOQ values (Page number 14).

Response 5: As per the suggestion of the reviewer, table 2 and table 3 were revised in the revised manuscript. With respect to table 3, all insignificant figures in the tables were corrected and the SD of all the factors were found within the significant ranges as suggested by the reviewer.

---

## [Editor Report · Decision Letter 2]

23 Dec 2019

Method Development and Validation for rapid identification of Epigallocatechin Gallate using Ultra-High Performance Liquid Chromatography

PONE-D-19-27318R2

Dear Dr. Sinha,

We are pleased to inform you that your manuscript has been judged scientifically suitable for publication and will be formally accepted for publication once it complies with all outstanding technical requirements.

With kind regards,

Marina Pinheiro

Academic Editor

PLOS ONE
---

## [Editor Report · Acceptance letter]

31 Dec 2019

PONE-D-19-27318R2 

Method Development and Validation for rapid identification of Epigallocatechin Gallate using Ultra-High Performance Liquid Chromatography 

Dear Dr. N:

I am pleased to inform you that your manuscript has been deemed suitable for publication in PLOS ONE. Congratulations! Your manuscript is now with our production department. 

With kind regards,

on behalf of

Dr. Marina Pinheiro 

Academic Editor

PLOS ONE